# Responses of Runoff and Soil Loss to Rainfall Regimes and Soil Conservation Measures on Cultivated Slopes in a Hilly Region of Northern China

**DOI:** 10.3390/ijerph18042102

**Published:** 2021-02-21

**Authors:** Haiyan Fang

**Affiliations:** 1Key Laboratory of Water Cycle and Related Land Surface Processes, Institute of Geographic Sciences and Natural Resources Research, Chinese Academy of Sciences, Beijing 100101, China; fanghy@igsnrr.ac.cn; Tel.: +86-010-6488-3172; 2College of Resources and Environment, University of Chinese Academy of Sciences, Beijing 100049, China

**Keywords:** cultivated land, rainfall regime, soil conservation measure, reduction efficiency, northern China

## Abstract

Cultivated land plays an important role in water and soil loss in earthy/rocky mountainous regions in northern China, however, its response to soil conservation measures and rainfall characteristics are still not fully understood. In the present study, 85 erosive rainfall events in 2011–2019 were grouped into three types, and the responses of runoff and soil loss to soil conservation measures and rainfall regimes on five cultivated plots with different slopes in the upstream catchment of the Miyun Reservoir were evaluated. Results found that mean event runoff depths and soil loss rates on the five plots ranged from 0.03 mm to 7.05 mm and from 0.37 t km^−2^ to 300.51 t km^−2^ respectively, depending on rainfall regimes, soil conservation measures, and slope gradients. The high frequency (i.e., 72.94%) rainfall regime A with a short rainfall duration (RD), low rainfall amount (P), and high mean rainfall intensity (I_m_) yielded a lower runoff depth and higher soil loss rate. Rainfall regime B with a longer RD, and a higher P and I_m_, however, produced higher a runoff depth and lower soil loss rate. Terraced plots had the highest runoff and soil loss reduction efficiencies of over 96.03%. Contour tillage had comparable sediment reduction efficiency to that of the terraced plots on gentle slopes (gradient less than 11.0%), while its runoff reduction efficiency was less than 13.11%. This study implies that in the Miyun Reservoir catchment and similar regions in the world, contour tillage should be promoted on gentle slopes, and the construction of terraced plots should be given ample consideration as they could greatly reduce water quantity and cause water shortages in downstream catchments.

## 1. Introduction

Soil erosion is an expansive environmental problem with significant ecological implications. It is associated with on-site land degradation, off-site sediment siltation in rivers, reservoirs, and lakes, and water resource use [1,2,3]. Thus, it directly and indirectly influences water, soil, and organism health as well as other earth surface processes [3].

It is widely recognized that land use affects soil loss. Among all the land use types, sloping cultivated land usually suffers higher soil loss and acts as an important sediment source area [4,5]. In the Ethiopian Plateau of Africa, the average soil loss rate (SLR) of sloping farmland is around 8000 t km^−2^ yr^−1^, representing the land use type suffering from the highest soil loss in sub-Saharan Africa [6]. In China, sloping farmland covers an area of around 24.5 million ha, occupying approximately a fifth of the cultivated land, and annual soil loss from sloping farmland is approximately 1.5 × 10^9^ t yr^−1^, contributing to around 30% of the total soil loss in China [7]. On the Loess Plateau, the SLR can be up to 9700–21,700 t km^−2^ yr^−1^, and under conditions of extreme rainstorms even 6.94 × 10^3^–95.89 × 10^3^ t km^−2^, which is dozens of times the rate of grass and forest lands [8]. In the Sichuan Basin of China, the SLR on sloping farmlands can also reach 3000–5000 t km^−2^ yr^−1^, and soil loss from the cultivated land accounts for 60–80% of the total [9]. Therefore, soil and water resource protection and corresponding scientific research on sloping farmland are particularly important and necessary.

The Miyun Reservoir is an important source of drinking water for people in Beijing, 70% of which is provided by the Miyun Reservoir [10,11,12,13]. Therefore, the water quality and quantity of the Miyun Reservoir and its upstream catchment have attracted much attention by local government and research scholars. In recent years, many studies have been conducted in this region, mainly focusing on water pollution [14], water purification [15], nitrogen and phosphorus loss [16,17], and soil pollution [11,18]. In the process of soil erosion, the dissolved pollutants in runoff and/or those adsorbed onto the sediments can be routed out, resulting in downstream water pollution [19]. Therefore, soil erosion in the upper reaches of the reservoir has also attracted attention, and a large number of soil conservation measures have been implemented in the mountainous areas, including contour tillage, terraces, tree planting, and dams in gullies and/or rivers [11,13,20].

In the last two decades, researches on soil erosion and its response to soil conservation measures have been done in this region. At the catchment scale, studies on the effect of land use change and soil conservation measures on runoff and sediment transport dynamics have been conducted [12,13,21,22,23]. At the slope scale, the characteristics of water and soil loss and their responses to soil conservation measures were also analyzed [19,21,24,25]. Nowadays, cultivated lands in the upstream catchment of the Miyun Reservoir still act as important sediment and pollution source areas, and contour tillage and terrace practices have been widely implemented in this region [12,13,21]. Intensive anthropogenic activities and lasting dry years have resulted in a dramatically reduced streamflow into the Miyun Reservoir [23,26]. Thus, comparisons of runoff and soil loss from cultivated lands in this region have important implications. However, evaluations of the responses of runoff and soil loss on cultivated lands to different rainfall characteristics and soil conservation measures are scarely done. Therefore, it is vitally important to cope with future extreme rainstorms through managing land use in the hilly regions around the capital city of China [13,23].

Therefore, runoff and soil erosion data from 85 erosive rainfall events over five cultivated plots were used to explore the responses of runoff and soil loss to soil conservation measures and rainfall characteristics in the upstream catchment of the Miyun Reservoir, northern China. The specific aims were to (i) compare the differences in surface runoff and soil loss under different soil conservation measures, (ii) identify their responses to different types of rainfall events, and (iii) give suggestions to implement reasonable soil conservation measures in the study area.

## 2. Materials and Methods

### 2.1. Study Site

The study area is located in the Shixia catchment (117°4′30″ E and 40°34′40″ N) (Figure 1), upstream of the Miyun Reservoir, some 90 km northeast of Beijing. The catchment lies in the Yanshan Mountains, and has a temperate territorial monsoon climate. Mean annual precipitation is 661.8 mm, ranging from 300 mm to 700 mm, 70% of which falls from June to August. The annual evaporation is 1840 mm.

The catchment has an area of 33 km^2^ with elevations of 130–390 m a.s.l. It exhibits an earth-rock hilly geomorphology. Around 20.6% of the catchment has a slope gradient od less than 8.7%, 50.2% of the slopes range from 8.7% to 26.8%, and slopes larger than 36.4% only account for 16.2% of the total. The lithology is dominated by gneiss, scattered with granite and limestone. The soil cover consists of cinnamon soils, as per the Chinese soil classification system which was developed on alluvial and diluvial parent materials. The soils are around 30 cm in depth, and around a 10 cm horizon of weathered rock mixed with soil appears under the soil layer. The catchment is mostly covered by artificial *Robinia pseudoacacia*, *Pinus tabulaeformis*, and economic forest. The major crops are corn (*Zea mays*) and wheat (*Triticum aestivum*).

### 2.2. Description of the Selected Runoff Plots

There are 22 runoff plots in the catchment (Figure 1). The boundaries of each plot were made of bricks and cement to prevent runoff from leaving or entering the plot. The plots are bare, covered with different types of vegetation, or implemented with different soil conservation measures. Among the 22 runoff plots, only five plots were cultivated, consisting of one plot without soil conservation measures, one plot with 4 m wide terrace, and three plots with contour tillage. In order to specifically study the effects of the soil conservation measures and slope gradients on the runoff and soil loss from cultivated slopes, all these five plots were selected in the present study. Each plot was 50 m^2^ in area and 5 m in width. The slope gradients range from 6.1% to 29.6% with soil thicknesses of around 30 cm. Detailed information of the plots was given in Table 1.

**Table 1 ijerph-18-02102-t001:** Description of the selected runoff plots.

No.	Gradient (%)	Length (m)	Width (m)	Area (m^2^)	Aspect	Type	Crop	Measure
1	29.6	10	5	50	Sunny	Cultivated	Corn	-
2	25.7	10	5	50	Northwestern	Cultivated	Corn	Contour tillage
3	6.1	10	5	50	Sunny	Cultivated	Corn	Terrace (4-m wide)
4	6.1	10	5	50	Sunny	Cultivated	Corn	Contour tillage
5	11.0	10	5	50	Sunny	Cultivated	Corn	Contour tillage

### 2.3. Data Collection

Surface runoff and soil loss were collected and measured after each rainfall event in 2011–2019. A nine-hole diversion bucket and a tank were used to collect the runoff mixed with sediment from the plots. The runoff amount was calculated and sampled with 1000 mL flasks after each rainfall. After settling for 24 h, sediment was separated from the water, dried in an oven at a temperature of 105 °C, and subsequently weighted to determine the sediment concentration. The soil loss of each plot was calculated by multiplying the average sediment concentration and the runoff amount. The runoff depth (H; mm) of the plot after each rainfall event was calculated by using the total runoff amount during an entire rainfall event period and plot area. An event SLR (t km^−2^ event^−1^) was obtained by using soil loss amount and plot area. Annual H (AH) and the annual SLR of each plot were obtained by summing the event-derived ones.

A self-recording rain gauge and rain barrel were installed near the plots to monitor rainfall process and rainfall amount. According to the records, rainfall duration (RD), rainfall amount (P), mean rainfall intensity (I_m_), and maximum intensities at 30 min (I_30_) and 60 min (I_60_) were obtained. Annual rainfall amount (AP) was obtained by summing event P.

Soils from the runoff plots were collected, and the soil texture and soil organic matter in Table 2 were obtained by Wang et al. [19]. The soil samples for moisture measurements before a rainfall event were taken from 0–10 cm and 10–20 cm depths using drilled soil cores. Sampling time intervals depended on the occurrence frequency of rainfall events. All the samples were transported to the laboratory, oven-dried, and weighed to determine their soil moistures.

### 2.4. Data Treatment and Statistical Analysis

In order to study the effect of rainfall characteristics on surface runoff and soil loss, the K-means clustering method was used to group the erosive rainfall events in 2011–2019. To determine the number of groups, many criteria were used. Normally, the classification must meet the ANOVA criterion of significance (*p* < 0.05). In the present study, rainfall eigenvalues of P, RD, I_m_, I_30_, and I_60_ were employed to group the erosive rainfall events.

In the present study, erosive rainfall was defined as the rainfall that induces runoff and soil loss. The mean annual runoff reduction efficiency (ARRE) of a soil conservation measure was calculated as follows:(1)$$\mathrm{ARRE}=\frac{{\mathrm{AH}}_{0}-{\mathrm{AH}}_{i}}{{\mathrm{AH}}_{0}}\;\left(i=2,3,4,\;\mathrm{or}\;5\right)$$
where AH_0_ represents the mean annual H on plot 1 (no soil conservation measures), and AH*_i_* represents the mean annual H on the *i*th plot (where a soil conservation measure is present). Therefore, separate ARRE for each plot were obtained. In order to evaluate the effects of contour tillage and terracing on surface runoff, their ARREs were not averaged in the present study. Similarly, the mean annual soil loss reduction efficiency (ASLRE) for each plot was calculated as follows:(2)$$\mathrm{ASLRE}=\frac{{\mathrm{ASLR}}_{0}-{\mathrm{ASLR}}_{i}}{{\mathrm{ASLR}}_{0}}\;\left(i=2,3,4,\;\mathrm{or}\;5\right)$$
where ASLR_0_ represents the mean annual SLR on plot 1, and ASLR*_i_* represents the mean annual SLR on the *i*th plot where a soil conservation measure is present. The calculated ASLREs were given the same treatment as the ARREs.

Pearson correlation analysis was performed to assess the relationships between H, SLR, and their influencing factors. Fisher’s protected least significant difference (LSD) test was conducted to compare the means of Hs and SLRs on the plots. Treatments were considered significant if *p* < 0.05.

## 3. Results

### 3.1. Rainfall Characteristics

In 2011–2019, the mean AP was 507 mm, ranging from 410 to 579 mm. In contrast, the mean annual erosive rainfall amount was 326 mm, ranging from 249 to 434 mm. There were 85 erosive rainfall events in the study period, ranging from seven to eleven rainfall events per year (Figure 2a). Most erosive rainfalls occurred in July and August (Figure 2b).

In comparison to I_30_ and I_60_, the I_m_ had the largest mean square value (i.e., 854.79; Table 3), resulting from the largest coefficient of variation (CV) for the 85 erosive rainfall events (Table 4). The separate eigenvalues of RD, P, and I_m_ were the significant rainfall eigenvalues used to group the 85 erosive rainfall events into three rainfall regimes (Table 3). Rainfall regime A occurred 62 times, accounting for 72.9% of the total. Rainfall regime B occurred 20 times, with an occurrence frequency that accounted for 23.5% of the total. Rainfall regime C had three rainfall events (Table 4). In accordance with the occurrence frequency, regime A had the shortest average RD of 176 min, the lowest average P of 26 mm, and the highest average I_m_ of 14.18 mm h^−1^. Inversely, rainfall regime C had the longest average RD, the highest average P, and the smallest average I_m_ of 3.30 mm h^−1^. These three rainfall regimes had comparable I_30_ and I_60_ values, without significant differences at the 0.05 level, due to their smaller CVs (Table 4). The CV varied from 0.10 to 1.31 for the five eigenvalues. The short RD, low P, and high I_m_ rainfall regime A had higher CV values for the three eigenvalues P, RD, and I_m_ than the other two rainfall regimes. With respect to the 85 erosive rainfall events, the average P, I_m_, I_30_, and I_60_ values were 33.51 mm, 11.45 mm h^−1^, 28.73 mm h^−1^, and 20.77 mm h^−1^, respectively. The average RD of the 85 erosive rainfall events was 354.77 min, ranging from 20 to 1940 min (Table 4).

### 3.2. Surface Runoff

During the study period, the mean AH differed greatly. Plot 1 without soil conservation measures had the highest AH ranging from 28.41 to 103.51 mm with an average of 71.30 mm. This value was larger than that of the contour tillage plot 2 which had a comparable slope gradient to plot 1. For the contour tillage plots 2, 4, and 5, the annual SLR increased with an increasing slope gradient. The AH on plot 2 ranged from 32.28 to 98.30 mm with an average of 63.07 mm, and the mean AH on plots 5 and 4 were 62.74 mm and 61.95 mm, respectively. The terraced plot 3 with a slope gradient of 6.1% yielded the least number of runoff events (i.e., 9; Figure 3b) and the lowest AH ranging from 0 to 11 mm with an average of 2.86 mm, which was significantly different from the results from the other plots at the 0.05 level (Figure 3a). For the five plots, event H was positively correlated with RD, P, I_30_, and I_60_ at the 0.01 or 0.05 levels (Table 5). However, H was not significantly correlated with I_m_ and antecedent soil moisture content (ASMC), although I_m_ was an efficient indicator for grouping the rainfall events (Table 3).

In comparison to the AH on plot 1 without soil conservation measures, 96.03% of surface runoff was reduced for plot 3. However, lesser runoff was intercepted by the contour tillage plots 2, 4, and 5, with ARREs of 11.52%, 13.11%, and 12.05%, respectively (Figure 4).

The number of runoff–soil loss events differed among the plots. On plot 1, 77 events occurred. Impacted by the soil conservation measures and slope gradient, 71, 67, 63, and 9 events occurred on plots 2, 5, 4, and 3, respectively. For each plot, more runoff–soil loss events were caused by rainfall regime A, followed by regimes B and C (Figure 3b). The average Hs of the 85 erosive rainfall events presented the same sequence (Figure 3c). Plot 1 yielded the highest H, and plot 3 yielded the lowest one. Rainfall regimes affected H on each plot. Under rainfall regime A, the Hs increased from 0.2 mm on plot 3, 6.19 mm on plot 4, 6.29 mm on plot 5, 6.35 mm on plot 2, to 7.03 mm on plot 1. Under rainfall regime B, the Hs increased from 0.23 mm on plot 3, 6.23 mm on plot 4, 6.44 mm on plot 5, 6.66 mm on plot 2, to 8.09 mm on plot 1. However, under rainfall regime C, plot 4 had the highest H of 16.4 mm and plot 3 had the lowest one of 2.94 mm. For each plot, rainfall regimes B and C had a higher H, and rainfall regime A had a lower H. For example, on plot 1, the Hs were 7.03 mm, 8.09 mm, and 14.42 mm under rainfall regimes A, B, and C, respectively. The average Hs on plots 1, 2, 4, and 5 under each rainfall regime were significantly higher than that on plot 3 at the 0.05 level (Figure 3c).

### 3.3. Soil Loss

Soil losses from the plots had almost the same changing patterns as their corresponding H levels at both the annual and event scales. The steepest plot 1 without soil conservation measures had the largest annual SLRs of 2838.11 t km^−2^. For the three contour tillage plots 2, 5, and 4, the mean annual SLRs increased with increasing slope gradient, ranging from 160.60 t km^−2^ on plot 4, 455.94 t km^−2^ on plot 5, to 2033.83 t km^−2^ on plot 2. The terraced plot 3 had the lowest mean annual SLR of 3.49 t km^−2^ on plot 3 (Figure 5a). Impacted by the implemented soil conservation measures and slope gradient, the mean annual SLRs on plots 1 and 2 were significantly different to those of the other plots at the 0.05 level. Similarly, the mean event SLRs increased from zero on plot 3, 7.00 on plot 4, 47.22 on plot 5, 215.35 on plot 2, to 300.51 t km^−2^ on plot 1. Similar to surface runoff, soil loss was also significantly correlated with P, I_30_, and I_60_, and was insignificantly correlated with RD, I_m_, and ASMC. In contrast to the mean annual SLR for plot 1, plot 3 had the highest ASLRE of 99.88%, and plots 4 and 5 also had a higher ASLRE with values of 94.34% and 84.29%, respectively. However, the ASLRE of plot 2 was only 28.34%.

In contrast, the mean event SLR under rainfall regime A was insignificantly higher than that of rainfall regime B for the individual plots (Figure 5b). However, the mean event SLR under regime C was higher than that of regimes A and B at the 0.05 level on each plot. For example, the SLRs on plot 1 were 319.54 t km^−2^, 225.39 t km^−2^, and 408.00 t km^−2^ under rainfall regimes A, B, and C, respectively.

### 3.4. Runoff–Soil Loss Relationship

Runoff and soil loss relationships for each plot under different rainfall regimes are shown in Figure 6. The scattered points were fitted with linear regressions. The slope b of the linear regression function y = a + bx reflects the sensitivity of the soil to erosion. Similarly, impacted by soil conservation measures and the slope gradient, the changing pattern of the b values agreed with those of the mean annual AH and SLR. For all the erosive rainfall events, the b value on plot 1 was the highest at 35.39, followed by values of 30.82 on plot 2, 9.04 on plot 5, and 1.94 on plot 4. However, there was no apparent relationship between runoff and soil loss on plot 3. For individual runoff plots, the b values were usually less under rainfall regime A, and higher under rainfall regimes B and C. For example, for plot 1, the b value was 25.16 under rainfall regime A, however the b values were 38.14 and 28.04 under regimes B and C, respectively.

## 4. Discussion

### 4.1. Effect of Antecedent Soil Moisture and Soil Crust

Soil moisture is an important factor in influencing surface runoff and soil loss, and higher ASMCs usually induce more runoff and soil loss [28]. However, there were no significant correlations between the ASMC, H, and SLR in the present study (Table 5). This could result from the interactions of the soil crust and rills as affected by ASMC and the runoff generation type in the study area. Under the impact of rainfall, an encrusted soil surface can greatly reduce the infiltration rate [29,30,31]. The presence of crust only 0.1 mm thick may reduce the infiltration rate from 800 cm day^−1^ to 70 cm day^−1^ [32]. Qinna and Awwad [33] found that the permeability of deep soils was up to 2000-fold higher than that of the surface soil crust. However, when rainfall intensity is high enough, the encrusted soil can be destroyed and rills and/or even ephemeral gullies develop that can increase the infiltration rate [34,35,36]. Soil crust is easily formed on the gentle slopes which increases surface runoff, and faster runoff velocity on steep slopes also increases surface runoff. This can explain the comparable ARREs for the gentle plots 4 and 5 and the steep plots 1 and 2. However, terracing can reduce runoff velocity and increase higher runoff infiltration, resulting in a higher ARRE for the terraced plot 3. In comparison to plots 4 and 5, less runoff generation events occurred on terraced plot 3. In respect of sediment loss control, the contour tillage on plots 4 and 5 can greatly filter sediment although it allows more runoff to run downslope. As a result, the ASLREs of plots 4 and 5 were just slightly lower than that of terraced plot 3. Further comparison indicated that the event SLRs from plot 5 ranged from 0–920 t km^−^^2^ event^−^^1^, whereas most of the events had an SLR of less than 100 t km^−^^2^ event^−^^1^. The event SLR on plot 4 ranged from 0 to 102 t km^−^^2^ event^−^^1^. Therefore, the ASLREs of plots 4 and 5 were also comparable.

Soils with a higher ASMC can accelerate the formation and destructive processes of soil crust [36], resulting in its complex effect on H and soil loss. Field investigations demonstrated that sometimes rills develop on the lower section of the plots after a high intensity rainfall event. These changes of soil crust and rills with rainfall duration make the relationship between ASMC and runoff complex [35]. The special runoff generation type in this region discussed below could also induce insignificant correlations of ASMC and H. Under short RD and high rainfall intensity conditions, the infiltration of excess runoff dominates, resulting in a lower H. However, longer RD rainfall and higher P events yielded a higher H (Figure 3b), due to the dominant saturation of excess runoff. Affected by the complex interactions of soil crust and the development of rills, and the changes in the runoff generation type with time duration, the ASMC did not significantly affect the H and SLR on the plots in the study area.

### 4.2. Effect of Rainfall Regimes

In the present study, H and SLR were not significantly correlated with I_m_, although it was an efficient indicator to group the rainfall events (Table 3), implying that I_30_ and I_60_ were more efficient than I_m_ in influencing runoff and soil loss [35,36,37,38]. In the study area, rainfall regime A is the most frequently occurring rainfall event, with a short RD, low P, and high I_m_. This type of rainfall regime has been proved to produce more surface runoff and soil loss than regimes B and C [38,39]. However, in the present study, rainfall regime A induced a lower H (Figure 3c). In the earth-rocky hilly region, the infiltration of excess runoff dominates the early stage of a rainfall event. However, as the duration of the rainfall event increases, the runoff generation type turns from infiltration excess runoff into saturation excess runoff [40]. In contrast to the infiltration excess runoff on the Chinese Loess Plateau [35,38,39], this kind of runoff generation type resulted in significantly positive correlations between RD and H (Table 5), inducing a higher H on the plots under rainfall regimes B and C (Figure 3b).

Consistent with previous studies [35,38,39], higher SLRs occurred under rainfall regime A (Figure 5b), resulting from its significantly higher I_m_ (Table 4). As mentioned above, soil crust usually develops under raindrop impact [30,35]. However, the encrusted soil which is developed on coarsely textured soil (Table 2) can easily be destroyed when rainfall event of higher intensity occurs, and more loose soil underneath is the readily available for erosion, resulting in a higher sediment concentration and SLR [36,41]. This inference can be verified from the fitting lines of H versus SLR under different rainfall regimes (Figure 6). For example, according to the regression functions of the fitting lines for plot 1 under rainfall regimes A and B, the average event H of 7.03 mm under rainfall regime A produced an SLR of 292.00 t km^−2^ event^−1^, which is larger than the SLR of 225.36 m^−2^ event^−1^ derived from the average event H of 8.09 mm under rainfall regime B. Rainfall regime C had a similar I_30_ and I_60_ to rainfall regime A, but with a much longer RD of 1171 min and a higher P of 96.37 mm (Table 4), resulting in the highest H and SLR (Figure 3c and Figure 5b). However, because rainfall regime C consisted of only three rainfall events, the majority of the soil loss from plots was still induced by rainfall regime A.

### 4.3. Effect of Slope Gradient

Slope gradient is an important factor in influencing surface runoff and soil loss [42,43,44]. As the slope gradient increases, the runoff-holding capacity of soil conservation measure will decrease. Zhao et al. [44] found that the efficiency of soil loss reduction by contour tillage decreased by 3.08% with a gradient increment of one degree. In the RUSLE2 model, the effect of slope gradient is regarded as the greatest influencing factor before contour failure [45]. In the present study, plots 2, 4, and 5 had the same soil conservation measure (i.e., contour tillage). Affected by slope gradients (Table 1), the average event Hs on the plots increased with increasing slope gradient, although their differences were insignificantly at the 0.05 level (Figure 3b). However, the average event SLR on plot 2 was significantly higher than that of plots 4 and 5 (Figure 5b). This result is consistent with the published literatures [46,47]. Therefore, the ASLREs of contour tillage measures on plots 4 (94.34%) and 5 (84.29%) with gentle slopes were rather higher than on plot 2, although their ARREs differed little, ranging from 11.52% to 13.11% for the 85 rainfall events. It has also been confirmed that contour tillage has a greater sediment reduction effect than runoff reduction effect by Jia et al. [7] through a meta-analysis in China.

### 4.4. Effect of Soil Conservation Measures

Contour tillage and terracing are the two most implemented soil conservation measures across China and the world [7,48,49]. In the present study, the selected runoff plots are nearby with the same soil type and rainfall characteristics. The primary differences lie in the implemented soil conservation measures and slope gradients. In the present study, the terraced plot 3 had a higher ARRE and a lower number of runoff generation events than the plots with contour tillage. This result agrees with the reported conclusions [50,51,52]. However, terraced plots had a slightly lower ASLRE to the contour tillage plots 4 and 5 (Figure 6). This could result from the intense sediment filter function of contour tillage measures on gentle slopes. In comparison, its filter function became lower on steep slopes (e.g., plot 2). These comparisons imply that contour tillage is efficient to control soil loss but cannot efficiently intercept runoff on gentle slopes.

In recent years, the deterioration in water quality and a shortage of water in the Miyun Reservoir have threatened Beijing’s drinking water supply [12,23], and thus “water saving” soil conservation measures should be promoted in this region [53]. In the present study, the terraced plot can effectively intercept both runoff and sediment that could greatly prevent runoff from entering into the Miyun Reservoir, affecting the amount of drinking water for Beijing. Furthermore, the construction of terracing is costly and it occupies a large part of a field. Therefore, terraces should be cautiously implemented in the study area. On gentle slopes, contour tillage should be promoted because it allows more runoff to route downstream whereas it can efficiently control soil loss.

## 5. Conclusions

In the present study, surface runoff and soil loss from five cultivated slopes were monitored in 2011–2019 in a hilly region of northern China, and 85 erosive rainfall events occurred during the study period. The 85 erosive rainfall events were grouped into three regimes based on K-means cluster classification method, and the effects of rainfall regimes and soil conservation measures (i.e., contour tillage and terracing) on runoff and soil loss of the cultivated plots were evaluated.

Responses of runoff and soil loss to the three rainfall regimes were different. The frequently occurring rainfall regime A had a short RD, low P, and high I_m_, producing a higher SLR and lower H. However, rainfall regime B with a higher RD, P, and low I_m_ yielded a higher H and lower SLR. Terraced plots can concurrently intercept runoff and sediment, resulting in the highest ARRE (i.e., 96.03%) and ASLRE (i.e., 99.88%) on gentle slopes. Although contour tillage measures had comparable ASLREs to terraced plot 3, their ARREs were much smaller than the terracing on gentle slopes. The effects of rainfall characteristics and soil conservation measures increased with decreasing slope gradients.

In the Miyun Reservoir catchment where water shortages exist, water-saving soil conservation measures should be given priority. Therefore, contour tillage should be promoted on gentle slopes, and terracing should be given further attention in both the study area and similar regions around the world, as it can intercept a large portion of surface runoff which could greatly reduce the downstream water quantity, which exacerbates water shortages and endanger the safety of drinking water in downstream areas.

## Figures and Tables

**Figure 1 ijerph-18-02102-f001:**
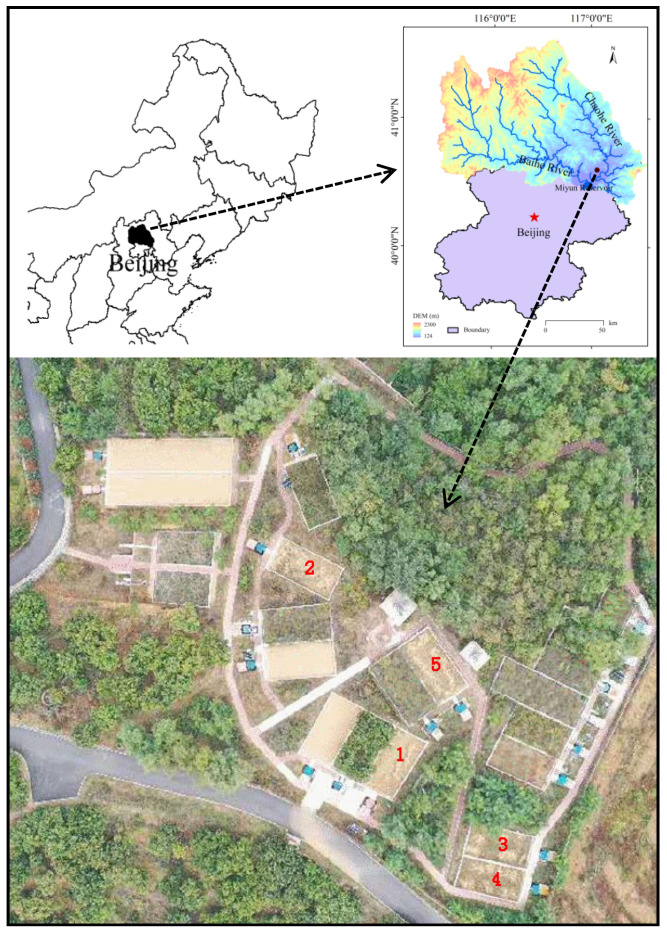
Location of the study area and the cultivated runoff plot selected. The red figures represent the plots in Table 1 (The bottom map showing the plots’ distribution was edited from Xu [27]).

**Figure 2 ijerph-18-02102-f002:**
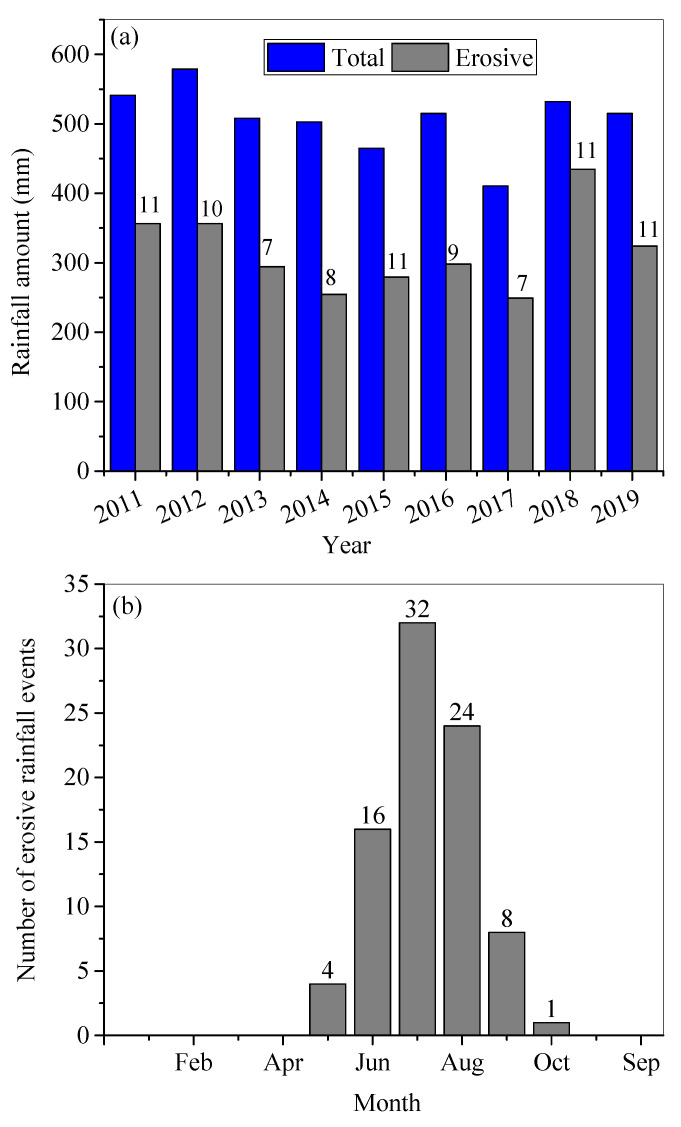
Annual total amount of rainfall and erosive rainfall between 2011–2019 (**a**). The number of erosive rainfall events in each month of the study period (**b**). The numbers on the erosive rainfall amount bars indicate the number of erosive rainfall events.

**Figure 3 ijerph-18-02102-f003:**
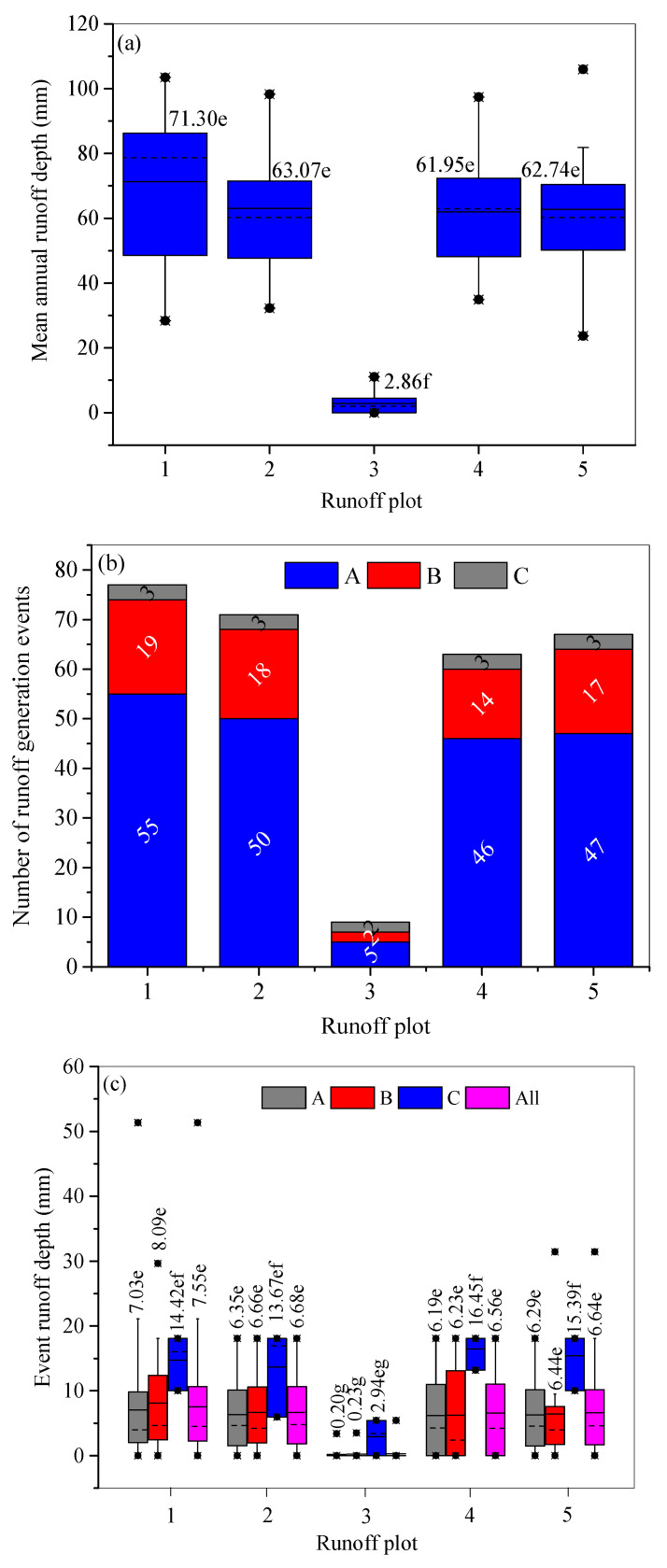
Mean annual runoff depth and standard deviation (**a**); number of runoff generation events induced by rainfall regimes A, B, and C on the five plots (**b**); and mean event runoff depth and standard deviation under each rainfall regime (**c**). Note: Values on the columns with the same letter in (**a**,**c**) were not significantly different at the 0.05 level. The solid line in the box represents the mean value, and the dotted line represents the median value.

**Figure 4 ijerph-18-02102-f004:**
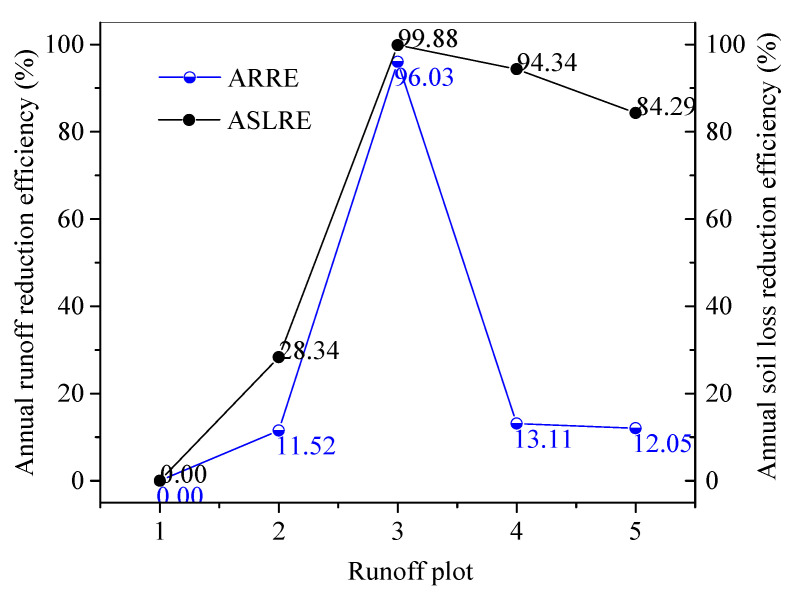
Annual runoff reduction efficiency and annual soil loss reduction efficiency of the soil conservation measures on the selected plots.

**Figure 5 ijerph-18-02102-f005:**
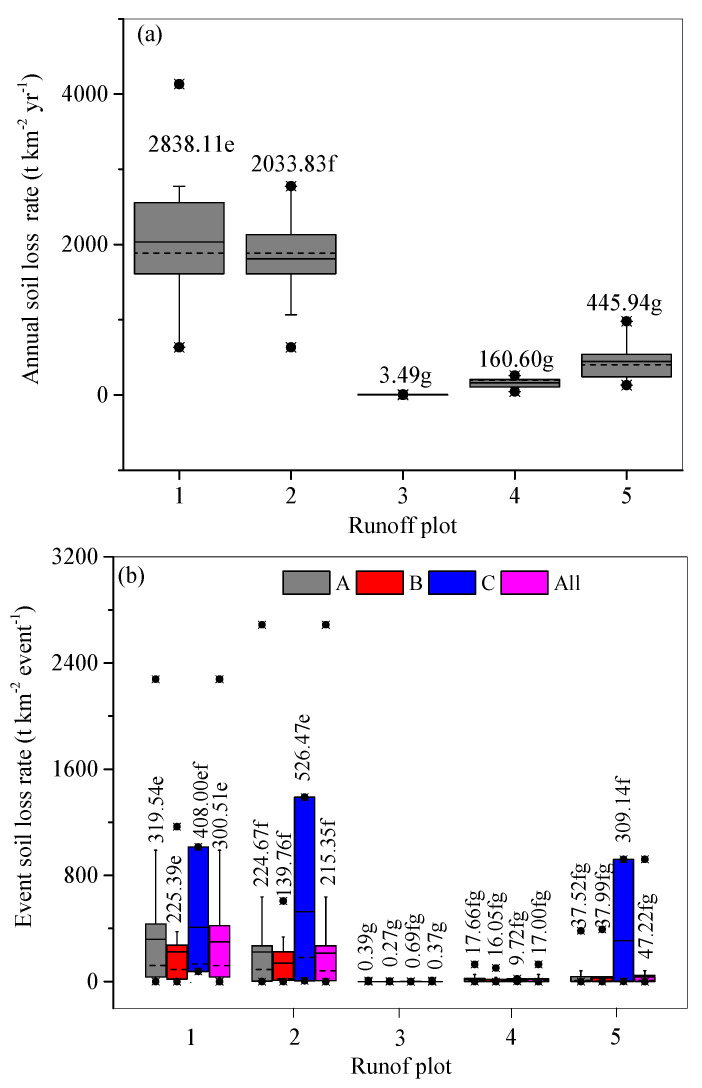
Annual soil loss rate (**a**) and event soil loss rate (**b**) for the 85 rainfall events under each rainfall regime. Note: Average values on the columns with the same letter are not significantly different at the 0.05 level. The solid line in the box represents the mean value, and the dotted line represents the median value.

**Figure 6 ijerph-18-02102-f006:**
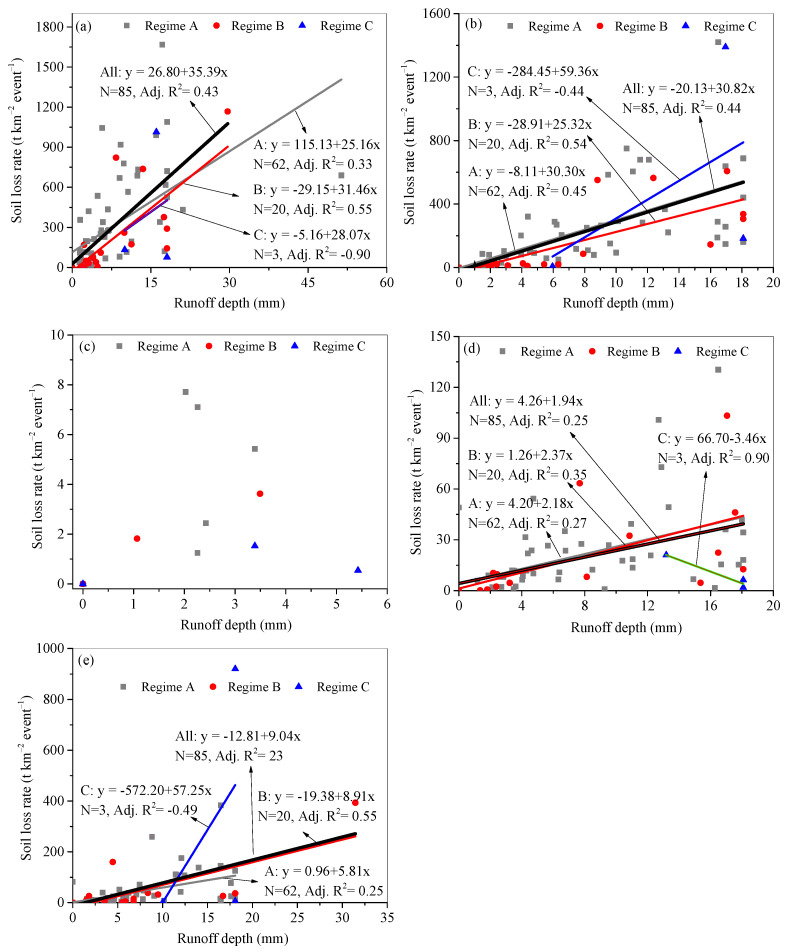
Relationships between the event runoff depth and event soil loss rate under different rainfall regimes for plot 1 (**a**), plot 2 (**b**), plot 3 (**c**), plot 4 (**d**), and plot 5 (**e**).

**Table 2 ijerph-18-02102-t002:** Soil properties of the runoff plots in the study area.

Eigenvalue	SOM (%)	Texture (Chinese Classification System)				
		>0.05	0.05–0.01	0.01–0.005	0.005–0.001	<0.001
Mean	1.57	60.9	15.00	3.22	6.00	14.88
St.D	0.69	12.58	4.36	1.53	1.00	2.55

**Table 3 ijerph-18-02102-t003:** One-way analysis of variance (ANOVA) results using a K-means clustering classification method.

Variables	Cluster		Error		F	Sig.
	Mean Square	df	Mean Square	df		
RD	4,980,828.25	2	23,349.50	82	213.32	0.00
P	9573.87	2	434.34	82	22.04	0.00
I_m_	854.79	2	121.57	82	7.03	0.00
I_30_	270.12	2	222.84	82	1.21	0.30
I_60_	59.44	2	162.97	82	0.37	0.70

**Table 4 ijerph-18-02102-t004:** Statistical features of the erosive rainfall events separated into three groups based on the rainfall regimes between 2011–2019.

Rainfall Regime	Eigenvalue	Mean	Range (Min–Max)	Standard Deviation	Variation of Coefficient	Number
A	P (mm)	26.01	4.80–105.20	16.52	0.64	62
	RD (min)	176.03	20.00–498.00	138.04	0.78	
	I_m_ (mm min^−1^)	14.18	2.07–69.6	12.68	0.89	
	I_30_ (mm min^−1^)	29.70	6.00–75.80	15.85	0.53	
	I_60_ (mm min^−1^)	21.17	3.60–69.60	12.76	0.60	
B	P (mm)	47.32	10.60–123.60	31.21	0.66	20
	RD (min)	705.30	500.00–1060.00	188.27	0.27	
	I_m_ (mm min^−1^)	4.20	1.03–11.36	2.84	0.68	
	I_30_ (mm min^−1^)	23.90	6.40–37.40	10.29	0.43	
	I_60_ (mm min^−1^)	18.74	4.50–57.20	12.44	0.66	
C	P (mm)	96.37	82.50–108.10	12.93	0.13	3
	RD (min)	1711.67	1580.00–1940.00	198.52	0.12	
	I_m_ (mm min^−1^)	3.30	3.07–3.74	0.34	0.10	
	I_30_ (mm min^−1^)	31.6	15.20–56.00	21.54	0.68	
	I_60_ (mm min^−1^)	23.77	14.40–42.00	15.79	0.66	
Total	P (mm)	33.51	4.80–123.60	25.53	0.76	85
	RD (min)	354.76	20.00–1940.00	376.01	1.06	
	I_m_ (mm min^−1^)	11.45	1.03–69.60	14.98	1.31	
	I_30_ (mm min^−1^)	28.40	6.00–75.80	14.95	0.53	
	I_60_ (mm min^−1^)	20.69	3.60–69.60	12.67	0.61	

Note: Rainfall regimes A, B, and C were the three types of rainfall events obtained from the K-means cluster analysis.

**Table 5 ijerph-18-02102-t005:** Pearson correlation coefficients between runoff, soil loss rate, and their influencing factors for the five runoff plots.

Runoff Plot	Eigenvalue	RD	P	I_m_	I_30_	I_60_	ASMC
1	H	0.224 *	0.607 **	0.098	0.528 **	0.487 **	−0.050
	SLR	−0.019	0.304 **	0.280 **	0.448 **	0.416 **	0.011
2	H	0.235 *	0.518 **	0.180	0.586 **	0.486 **	−0.060
	SLR	0.132	0.349 **	0.186	0.404 **	0.381 **	−0.096
3	H	0.402 **	0.521 **	0.024	0.286 **	0.284 **	0.002
	SLR	−0.004	0.282 **	0.127	0.245 *	0.320 **	0.049
4	H	0.316 **	0.581 **	0.061	0.514 **	0.441 **	0.003
	SLR	−0.100	0.219 *	0.181	0.281 **	0.287 **	0.112
5	H	0.220 *	0.521 **	0.182	0.542 **	0.491 **	0.006
	SLR	0.315 **	0.426 **	0.070	0.321 **	0.345 **	0.134

Note: SLR: soil loss rate; ASMC: antecedent soil moisture content. * represents significance at the 0.05 level, and ** represents significance at the 0.01 level.

## Data Availability

The data presented in this study are available on request from the corresponding author.

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
