# Peer review of "Responses of Runoff and Soil Loss to Rainfall Regimes and Soil Conservation Measures on Cultivated Slopes in a Hilly Region of Northern China"

_ijerph, 2021, doi:10.3390/ijerph18042102_

Round 1

Reviewer 1 Report

Summary

Some behavior has been mentioned which is related to rainfall event and soil erosion with detail matters. The subjection has important soil conservation for future.

Finally, I could say to you that this paper is effective to your Journal paper.

Question

Figure 1 is NOT clear, please prepare clear phot and illustration.

Line 90 through 94

It is better to apply more information on investigated area.

Line 138 through 149

Two equations are indicated that they are original or refer from other papers ?

Section 3.3

Soil loss are considered, and I have question as following: What kind od soil in loss ?

Section 3.4

It is sure that soil loss have significant to related to slope incline or land shape. Pleases would you investigate to land shape at investigation area.

Author Response

Dear reviewer, Thanks for your work. The responses to the comments or suggestions were listed below. 1, Figure 1 is NOT clear, please prepare clear phot and illustration. Response: Thanks. A new photo was used in the revised manuscript. The photo was cited from a thesis paper by Xu (2020) which has been added in the reference list. 2, Line 90 through 94, It is better to apply more information on investigated area. Response: Yes. More information, including slope degree, soil classification, evaporation were added in the area description. 3, Line 138 through 149, two equations are indicated that they are original or refer from other papers ? Response: Yes, they are original, not from other papers. 4, Soil loss are considered, and I have question as following: What kind of soil in loss ? Response: The soil cover consists of Cinnamon soils in Chinese soil classification system which is sensitive to soil crusting. 5, It is sure that soil loss have significant to related to slope incline or land shape. Pleases would you investigate to land shape at investigation area. Response; It is a good question. In the study area, there are slope incline. However, the runoff plots are straight. Therefore, slope incline was not considered in the present study.

Reviewer 2 Report

Dear Authors,

I think that the manuscript needs quite a bit of reorganization, must be tightened, make some parts clearer, give more precise captions for figures and X and Y axis titles, etc. There are a lot of minor issues that grow into a need for a big “major revision”.

Some details:

It would be nice to know a little bit more about the soils investigated.

All soils from all plots were the same?

Is there any other info about the soils than a SOM average and a PSD %??? And they were approximatey 30 cm thick.

If there is basically gneiss with some limestone and granite, what was underneath your soil?

Is there hard rock underneath the 30 cm soil, or there is something else between your soil and rock materials??? If so, how thick?

Table 3 and 4. I feel like there were more to explain about the results, e.g. what about the CV values? Do they explain anything? etc-etc.

What about the significant and non-significant differences? Why I30 and I60 did not produce significant differences while the overall mean did???? What is the reason for these results?

Figure 3 and its presentation is a mess! Please make it more clear! Also, all figures must be understandable as stand alone, so you cannot use abbreviations on figures, no matter if you explained these abbreviations in the materials and methods part.

Why do you give soil loss values in t/km2? Why not in t/ha?

Why do you have 1 control (without mitigation), 1 terraced and besides that, 3 contour tilled plots???

You explain ARRE and ASLRE in lines 138 to 143 but later on you do nothing with them.

There are also English parts that are not clear.

More comments in the attached pdf file.

I cannot review your discussion and conclusion until you do not clarify your methods and results.

Regards, Reviewer X

Author Response

Dear reviewer,

Great thanks for your invaluable comments, according to your comments, responses were made for each comments. Furthermore, the manuscript was also checked and some other errors were also corrected. Specific responses to the comments were listed below.

1, I think that the manuscript needs quite a bit of reorganization, must be tightened, make some parts clearer, give more precise captions for figures and X and Y axis titles, etc. There are a lot of minor issues that grow into a need for a big “major revision”.

Response: Yes, great thanks for you invaluable comments or suggestions. The specific responses were given below.

2, It would be nice to know a little bit more about the soils investigated. All soils from all plots were the same? Is there any other info about the soils than a SOM average and a PSD %??? And they were approximatey 30 cm thick.

Response: Yes, all the soils in the plots belong to one type of soil, i.e., they belong to Cinnamon soils in Chinese soil classification system. More soil data was not available, except for SOM and PSD%. Yes, they are around 30-cm thick.

3, If there is basically gneiss with some limestone and granite, what was underneath your soil? Is there hard rock underneath the 30 cm soil, or there is something else between your soil and rock materials??? If so, how thick?

Response: The weathered rock mass mixed with soil appeared underneath the soils. A weathered rock with soil horizon is about 10-cm depth under the soils. The information was added in the revised manuscript.

4, Table 3 and 4. I feel like there were more to explain about the results, e.g. what about the CV values? Do they explain anything? etc-etc.

Response: The CV values of P, RD, and Im were given, and their meanings were also given. Furthermore, comparisons were done for the CV values among the rainfall regimes A, B, and C. This information was given in the revised manuscript.

5, What about the significant and non-significant differences? Why I30 and I60 did not produce significant differences while the overall mean did???? What is the reason for these results?

Response: The difference is a statistical issue. In SPSS software, initial cluster centers were first determined casually, the distance of each data point to the center was then calculated for each eigenvalue. Therefore, the distances of each eigenvalue were compared using F statistics and then significant differences or not were determined for each group. In the present study, Im, I30, and I60 were correlated. Therefore, one eigenvalue of rainfall intensity should be selected from Im, I30, and I60. The variation of Im that was expressed as coefficient of variation (CV) was the largest among the three rainfall intensity variables (i.e., Im, I30 and I60). Therefore, Im can significantly differetiate the rainfall events, whereas I30 and I60 did not. Some information was added in the revised manuscript. However, specific explanation for the theory of K-mean cluster analysis in SPSS was not given in the present study because it is beyond of this study. 

6, Figure 3 and its presentation is a mess! Please make it more clear! Also, all figures must be understandable as stand alone, so you cannot use abbreviations on figures, no matter if you explained these abbreviations in the materials and methods part.

Response: Thanks. This figure and its presentation were remade, and the full names were used in the revised manuscript.

7, Why do you give soil loss values in t/km2? Why not in t/ha?

Response: Yes, h/ha was used in some papers, however, t/km2 was also commonly used, especially in China. Therefore, this unit was not changed in the revised manuscript.

8, Why do you have 1 control (without mitigation), 1 terraced and besides that, 3 contour tilled plots???

Response: Because there are only five cultivated plots in the study area. The explanation was given in the revised manuscript.

9, You explain ARRE and ASLRE in lines 138 to 143 but later on you do nothing with them. There are also English parts that are not clear.

Response: Later, they appeared at the 4.4 section, and some comparisons were also given using these two abbreviated words.

10, More comments in the attached pdf file. I cannot review your discussion and conclusion until you do not clarify your methods and results.

Response: Great thanks for your careful checks for my manuscript. All the suggestions or comments provided in the .pdf file were carefully considered, and revisions were made. Furthermore, careful check was also done for the discussion and conclusion sections.

10.1 More explanation is needed why do you have  3 plots with contour tillage if you have only on plot with terrace and one plot with ni measures!

Response: Among the 22 runoff plots, only five plots were cultivated. This information was given in the revised manuscript.

10.2 what is the difference between semi-shady and semi-sunny???

Response: Yes, they could not easily understood. Therefore, they replaced by “Northwestern” and “Southeastern” in the revised manuscript.

10.3 in 2.3 section, “Over” should be “oven”.

10.4 it would be much easier to tell (0-10 and 10-20 cm depth)

Response: Yes, this expression was edited as “…. were taken from 0-10 cm and 10-20 cm depth using drilled soil cores” in the revised manuscript.

10.5 I do not understand. If it was runoff, it is not soil moisture content. I do not understand the relation of the subtratcion of fresh soil and runoff????!!!! Please clarify!

Response: Yes. It is dry method using oven in fact. Therefore, new expression was used: oven-dried and wei weighted to determine their soil moistures.

10.6 For table 1, according to what classification? No SD, why?

Response: It is Chinese soil classification system, this information was given. SD was forgotten to be added. It was given in the revised manuscript.

10.7 For the expression “Event-averaged”,

Response: It indicates “mean event”, to avoid confusion, this expression was replaced by “mean event”.

10.8 it is explained that i=2,3,4, or 5 but it would be nice to know what happened exactly during the calculations, all of them were used and than averaged or contour and terraced were calculated separately and if so, how did you deal with having 3 contour and one terrace?

Similarly in case of ARRE!

Response: Yes, the reduction efficiency was calculated for each plot using annual averages, i.e., it was calculated separately for each plot, and the efficiencies of the three contour tillage plots were not averaged. This information was added in the revised manuscript. The result has been drawn in Figure 6.

10.8 In the caption of Figure 2, two words “in” and “monthly”

Response: “in” was replaced by “in between”. In the captions, Figure 2b means the monthly distribution of the erosive rainfall events. In order to avoid misunderstanding, the sentence was rewritten as “the numbers of the erosive rainfall events in each month in the study period”.

10.9 t is not 'and' as these are two separate issues!!! so officially it is the statistical features of the erosive events separated into 3 groups based on the rainfall regimes!

Response: Yes, “in” was replaced by “in between”, and according to the suggestion, the sentence was rewritten as “statistical features of the erosive events separated into 3 groups based on the rainfall regimes”.

10.10 For figure 3, the comments “If it was mean, why do you have the marks for other values? And, if you mark a range, why don't you use a boxplot?”; Please improve quality of the figure. It is really pixelized. For figure 3b, “If it was event H and it was expressed in mm event-1, why do you mark a range? Then it is average event H, or am I mistaken? From the text it looks like I am all right, as there you already write average!”

Response: Yes, a boxplot was used in the revised manuscript. The abbreviation of AH were replaced by its full name. The poor quality could be caused by the changes in different file types. To avoid misunderstanding, figure 3b was also replaced by a boxplot, and the explanation for Y-axis was also changed as event runoff depth. The letters used to explain the difference on the bars were also replaced by “e,f, and g” to replace “a and b”.

Similarly, all the abbreviated words were replaced by their full names for the figures bellow (Figures 4-6).

10.11 Please check English! E.g. "on" is missing before the first "plot", this way it is not understandable! On the other hand, why 1,71,67 and 63? Why not in order? Is ther any reason of the order? If so, please tell us here in the text!!!

Also in the second case, why 2,5,4,3???

Response: Response: Yes. Here “on” is missing. Among the figures “1, and 71, 67, 63, and 9”, the figure “1” after “plot “, it is plot 1, not the numbers of erosive rainfall events on the plots. Therefore, “71, 67, 63, and 9” is in order. In order to avoid misunderstanding, two words “in contrast” were added between “1” and “and 77…”. Because the sequence of the numbers of erosive rainfall events in the study period is impossible to agree with that of plot number, the plot number of 2, 5, 4, and 3 is not in order.

10.12 I do not understand this from-to relation as the numbers after the word "from" are bigger than after the word "to", so we can not talk about an increase in this case!

Response: Yes, it is not good. According to this suggestion, the “from … to ” sentences were rewritten in the revised manuscript.

10.13 English expressions at the end of section 3.3 and at the first paragraph of section 3.4 section were also corrected.

10.14 For the discussion and conclusions, carefully check was also made.

Except for the comments or suggestions mentioned, after careful check, other errors were also edited or corrected in the revised manuscript.

Round 2

Reviewer 2 Report

Dear Authors,

Thank you for the clarification.

I still have some concerns, now that I could reach the Discussion and Conclusion part, I found some of your results there, so I ask you to re-organize your manuscript, and put all your results in the results chapter.

I have major concerns about the runoff depth.

Other than that, some answers and more comments:

8, Why do you have 1 control (without mitigation), 1 terraced and besides that, 3 contour tilled plots???

Response: Because there are only five cultivated plots in the study area. The explanation was given in the revised manuscript.

I understand but why did you need 3 contour plots when you had one control and one terraced for comparison? Why did you need 3 versus 1 and 1, I mean, what was the reason „statistical analyses-wise”? Having these free under cultivation out of your 22 is not explaining the numbers!

9, You explain ARRE and ASLRE in lines 138 to 143 but later on you do nothing with them. There are also English parts that are not clear.

Response: Later, they appeared at the 4.4 section, and some comparisons were also given using these two abbreviated words.

OK, I see. It is very strange that you introduce these two methods in your materials and methods part but you only use the results in the Discussion chapter. I think, if you have results, you need to include them in the Results chapter and discuss your results in the discussion chapter.

Actually, it is also to the case with Table 5. It belongs to your results chapter.

You only discuss the effects of slope gradient in your discussion chapter, regardless of the fact that it must have a huge impact on your results, it should be there.

10.2 what is the difference between semi-shady and semi-sunny???

Response: Yes, they could not easily understood. Therefore, they replaced by “Northwestern” and “Southeastern” in the revised manuscript.

I only found Northwestern. Southeastern is still Sunny.

10.8 it is explained that i=2,3,4, or 5 but it would be nice to know what happened exactly during the calculations, all of them were used and than averaged or contour and terraced were calculated separately and if so, how did you deal with having 3 contour and one terrace?

Similarly in case of ARRE!

Response: Yes, the reduction efficiency was calculated for each plot using annual averages, i.e., it was calculated separately for each plot, and the efficiencies of the three contour tillage plots were not averaged. This information was added in the revised manuscript. The result has been drawn in Figure 6.

Again, as you said, the results has been drawn. If these are results, why are they not in the results chapter???

Furthermore: I am sorry, I do not believe that ARRE is basically the same on a 25.7% slope and on a 6.1 % slope. How dou you explain that on plot 3, where the number of runoff generation events is below 10, while in case of plot 4 and 5 it is above 10 and their ARRE is so different while their ASLRE is so similar?

Especially that the soil loss rate in case of Plot 4 is between 0 and 35  while in Plot 5 it is between 0 and 500 t/km2/event????????

I also attach the pdf file with some comments, some was copied from there.

Regards, Reviewer X

Author Response

Dear reviewer,

Great thanks for your careful review and valuable suggestions or comments for this manuscript. According to your comments or suggestions, the manuscript was carefully checked, and point by point response was given below.

1, Why do you have 1 control (without mitigation), 1 terraced and besides that, 3 contour tilled plots???

Response: Now, I understand you question. Because only five plots are cultivated, and slope gradient is also an important factor influencing runoff and soil loss from cultivated, therefore, all the three contour tillage plots were selected. This information was given in the 2.2 section of the revised manuscript.

2, I only found Northwestern. Southeastern is still Sunny.

Response: Yes, from the photo, it also belongs to sunny slope, because it just a little deviation from the south. Therefore, it was corrected.

3, For the ARRE and ASLRE, the comments were: It is very strange that you introduce these two methods in your materials and methods part but you only use the results in the Discussion chapter. I think, if you have results, you need to include them in the Results chapter and discuss your results in the discussion chapter.

Actually, it is also to the case with Table 5. It belongs to your results chapter. You only discuss the effects of slope gradient in your discussion chapter, regardless of the fact that it must have a huge impact on your results, it should be there.

Response: I understand now. Thanks for the valuable suggestion.  ARRE and ASLRE results, and Table 5 were divided into two parts that were distributed in the 3.2 and 3.3 sections. The effects of slope gradient on runoff, soil erosion, and runoff-soil loss relationships were also separately added into 3.2, 3.3, and 3.4 sections.

 4, It is absolutely misleading! The runoff depth is depending on the time, that the runoff spent on the plot. If you only consider the total runoff amount and plot area, it does not give you the runoff depth. E.g. if there was 100 mm runoff in 100 minutes and 100 mm runoff in 10 minutes it is exactly 10 times more H value!

Response: In the present study, runoff depth is the amount of runoff per unit area during the entire rainfall period. In fact, it has nothing to do with time. For example, one cubic meter of water in the runoff tank is collected after a rainfall event, and the runoff plot area is 20 square meters, then the runoff depth is 5 cm, regardless of whether the rainfall lasts for 20 minutes or 100 minutes. On the contrary, if time is considered, it is called “runoff rate” with a unit of mm min-1, not runoff depth. In order to verify this concept, I also read some published papers within which, the runoff depth was also calculated using total runoff amount to divide runoff plot area, not considering the time. Therefore, in order to avoid misleading, the information “for the entire rainfall period” was added after “The runoff depth (H; mm) of runoff plot after each rainfall event was calculated by using total runoff amount” in 2.3 data collection.

Three exampled papers:

Guo, M.M., Wang, W.L., Li, J.M., Bai, Y., Kang, H.L., Yang, B. 2020. Runoff characteristics and soil erosion dynamic processes on fourtypical engineered landforms of coalfields: An in-situ simulated rainfall experimental study. Geomorphology 349, 106896.

Liang, Y., Jiao, J.Y., Tang, B.Z., Cao, B.T., Li, H., 2020. Response of runoff and soil erosion to erosive rainstorm events and vegetation restoration on abandoned slope farmland in the Loess Plateau region, China. Journal of Hydrology, 584, 124694.

Puntenney-Desmond, K.C., Bladon, K.D., Silins, U., 2020. Runoffand sediment production from harvested hillslopes and the riparianarea during high intensity rainfall events. Journal of Hydrology, 582, 124452.

5, Wei weight, and .. in the pdf file.

Response: Sorry, they were wrongly written. They were corrected in the revised manuscript.

6, Response: “erosive” was added after “85” in the revised manuscript.

7, variance of what????

Response: It is a kind of statistical analysis. To avoid misunderstanding, “one-way” was added before “analysis of variance”.

8, regimes must be explained somewhere, below the table or in the caption but readers cannot be expected to roll back and forth to find out info about the regimes!

Response: Thanks. One note was given in the caption of Table 4.

9, according to fig3b, there was only 3 events with regime A!!!!!!!!!!

Response: Thanks for your careful check. I am sorry. It is an error of the legend. The legend was corrected in the revised manuscript.

10, I am sorry, I do not believe that ARRE is basically the same on a 25.7% slope and on a 6.1 % slope.

Response: Yes, it seems strange. I checked the event data for plots 2 (25.7% slope) and 4 (6.1% slope). I found that the event runoff depths of plots 2 and 4 were similar for the 85 erosive rainfall events. The monitored result can be explained by the encrust soil during rainfall which is easily formed on gentle slopes under rainfall splash. The encrusted soil can greatly reduce runoff infiltration. On the contrary, soil crust is not easily formed on steep plot although runoff velocity is faster on the steep slope. The interaction of runoff velocity and the effect of soil crust on runoff infiltration could explain the similar runoff depth on the 27.9% and 6.1% plots, although directly monitored data of soil crust was not available in the present study. This phenomenon was discussed and added at the 4.1 section in the revised manuscript.

11, How do you explain that on plot 3, where the number of runoff generation events is below 10, while in case of plot 4 and 5 it is above 10 and their ARRE is so different while their ASLRE is so similar?

Especially that the soil loss rate in case of Plot 4 is between 0 and 35 or while in Plot 5 it is between 0 and 500 t/km2/event????????

Response: In comparison to plots 4 and 5, less runoff generation events occurred on plot 3, this means terrace can reduce runoff velocity, and increase higher runoff infiltration, resulting in higher ARRE on plot 3. In respect of sediment loss control, contour tillage on plots 4 and 5 can greatly filter sediment although it allows more runoff to run downslope. By this way, the ASLREs of plots 4 and 5 were just slightly lower than that of terraced plot 3. This information was added in the revised manuscript.

The data was checked, the event SLRs from plot 5 ranged from 0-920 t km-2 event-1, however, most of the events had SLR less than 100 t km-2 event-1. The event SLR on plot 4 ranged from 0 to 102 t km-2 event-1. Therefore, their ASLREs were comparable. This phenomenon was discussed and added at the 4.1 section in the revised manuscript.

12, it is explained that i=2,3,4, or 5 but it would be nice to know what happened exactly during the calculations, all of them were used and than averaged or contour and terraced were calculated separately and if so, how did you deal with having 3 contour and one terrace? Similarly in case of ARRE!

Response: Yes, the reduction efficiency was calculated for each plot using annual averages of runoff and soil loss rate, i.e., it was calculated separately for each plot, and the runoff and soil loss reduction efficiencies of the three contour tillage plots were not averaged (The treatment of the calculated ARREs and ASLREs can also be shown in Figure 6). This information was added in the 2.3 section.